

# Flux balance modelling to predict bacterial survival during pulsed activity events

Nicholas A. Jose[1], Rebecca Lau[1], Tami L. Swenson[1], Niels Klitgord[1], Ferran Garcia-Pichel[2], Benjamin P. Bowen[1], Richard Baran[1], Trent R. Northen[1]

5    [1]Environmental Genomics and Systems Biology Division, Lawrence Berkeley National Laboratory, Berkeley, California 94720, United States

[2]School of Life Sciences, Arizona State University, Tempe, Arizona 85287, United States

*Correspondence to*: Trent Northen (TRNorthen@lbl.gov)



**Abstract**

Desert biological soil crusts (BSCs) are cyanobacteria-dominated, surface soil microbial communities common to plant interspaces in arid environments. The capability to significantly dampen their metabolism allows them to exist for extended periods in a desiccated dormant state that is highly robust to environmental stresses. However, within minutes of wetting, metabolic functions reboot, maximizing activity during infrequent permissive periods. *Microcoleus vaginatus*, a primary producer within the crust ecosystem and an early colonizer, initiates crust formation by binding particles in the upper layer of soil via exopolysaccharides, making microbial dominated biological soil crusts highly dependent on the viability of this organism. Previous studies have suggested that biopolymers play a central role in the survival of this organism by powering resuscitation, rapidly forming compatible solutes and fuelling metabolic activity in dark, hydrated conditions. To elucidate the mechanism of this phenomenon and provide a basis for future modelling of BSCs, we developed a manually-curated, genome-scale metabolic model of *Microcoleus vaginatus* (iNJ1153). To validate this model, GC/MS and LC/MS were used to characterize the rate of biopolymer accumulation and depletion in in hydrated *Microcoleus vaginatus* under light and dark conditions. Constraint-based flux balance analysis showed agreement between model predictions and experimental reaction fluxes. A significant amount of consumed carbon and light energy is invested into storage molecules glycogen and polyphosphate, while β-polyhydroxybutyrate may function as a secondary resource. Pseudo-steady state modelling suggests that glycogen, the primary carbon source with the fastest depletion rate, will be exhausted if *M. vaginatus* experiences dark wetting events four times longer than light wetting events.

**1 Introduction**

BSCs, estimated to contain ~4.9 Pg of terrestrial carbon globally, play critical roles in stabilizing the soil in desert and arid lands that comprise nearly 40% of planetary dry land masses (Garcia-Pichel et al., 2002;Elbert et al., 2012). BSCs exist in a desiccated and metabolically dormant state that is highly robust to environmental stresses, yet are able to rapidly re-boot metabolism within minutes of wetting enabling them to capitalize on infrequent pulsed activity events that occur upon wetting (Garcia-Pichel et al., 2013). The scale and potential sensitivity of BSCs to temperature and wetting frequency/duration make them particularly relevant to understanding the impact of climate change on soil microbial communities (Grote et al., 2010). Climate models predict alterations in precipitation frequency, intensity and seasonality all of which may negatively impact BSCs given that they often exist on the fringe of habitability (Belnap et al., 2004;Johnson et al., 2012). The validity of this concern has been experimentally validated by Reed et al., who showed a demise of mosses from BSCs when exposed to frequent low-intensity wetting events (Reed et al., 2012).

Pioneering cyanobacteria, such as *Microcoleus vaginatus*, initiate the formation of BSCs by binding particles in the upper layers of the soil with polysaccharide fibers (Garcia-Pichel and Wojciechowski, 2009). This not only stabilizes the soil, but creates a local environment that is conducive to diverse succession of microbes. However, development of mature



crusts from raw soils is a slow process, typically taking many years and presenting many challenges to environmental restoration efforts. The reason for this phenomenon is not well understood, but has been ascribed to the limited and sporadic periods of growth. Presumably BSC organisms have evolved to capitalize on wetting events that range from several hours to a few days, and appear to be poised to immediately restart metabolism (Potts, 1999). Indeed, after re-hydration, BSC

microorganisms start respiration within seconds, photosynthesis within minutes, and nitrogen fixation within tens of minutes (Rajeev et al., 2013;Garcia-Pichel and Belnap, 1996). Somewhat paradoxically, isolates of *M. vaginatus* grown in idealized laboratory conditions are still extremely slow-growing with doubling times of weeks. This may be a result of ingrained evolutionary pressures that prioritize robustness to various environmental stresses over growth.

      Previous studies on the desiccation cycle in *M. vaginatus* based on transcript analysis in intact BSCs indicated

multiple metabolic states (Rajeev et al., 2013). These states are associated with genes responsible for biopolymer formation and depletion, such as cyanophycin, glycogen, β-polyhydroxybutyrate (PHB), and polyphosphate. Wetting of dry crusts results in increased expression in genes for sugar transporters and biopolymer biosynthesis. During the Diel cycle, the expression patterns of biopolymer-associated genes suggest that depletion occurs at night and accumulation during the day. Drying resulted in elevated expression of genes involved in biopolymer metabolism, including glycogen breakdown.

Together these observations reinforce the centrality of osmolytes and biopolymers in the *M. vaginatus* desiccation cycle. Consistent with this view, our recent $D_2O$ labelling study of wetted *M. vaginatus* and *Synechococcus sp.* 7002 revealed that in light, low-growth, constant osmotic conditions many osmolytes are rapidly turned over (Baran et al., 2017). To gain insights into this process, stable isotope probing experiments were performed using the osmolyte glucosylglycerol, showing that *Synechococcus sp.* 7002 rapidly converts this osmolyte into glycogen. This suggests a mechanism where high fluxes of

compatible solutes are maintained through conversion to biopolymers. This high flux would enable cells to survive rapid desiccation by simultaneously turning off compatible solute polymerization and mobilizing biopolymer hydrolysis. Therefore, we expect wet-up, dry-down and dark reactions likely have fixed biopolymer costs whereas the light reactions enable replenishment of biopolymer reserves. Given the importance of biopolymers to *M. vaginatus*, the balance of costs vs. accumulation could be used to predict net carbon accumulation and survival of *M. vaginatus* during various climate

scenarios.

      Here, as a first step towards this goal, we constructed a genome-scale metabolic network of *M. vaginatus* (iNJ1153). To our knowledge, this is the first for a terrestrial cyanobacteria and describes a flux balance approach that accounts for biopolymer accumulation using polyphosphate, β-polyhydroxybutyrate (PHB), and glycogen, important storage molecules and osmolytes (Diamond et al., 2015). We use this model in combination with direct measurements of biopolymer

concentrations and carbon dioxide flux to examine dark and light flux distributions in *M. vaginatus*. We interpret these results using a simple cost/benefits framework for biopolymer depletion in the dark 'costs' and accumulation in the light 'benefits'.



## 2 Materials and Methods

### 2.1 Chemicals

Potassium phosphate (dibasic) (CAS 7758-11-4), crotonic acid (CAS 107-93-7), D-glucose (CAS 50-99-7), 98% methoxyamine hydrochloride (CAS 593-56-6), pyridine (CAS 110-86-1), fatty acid methyl ester (FAME) standards (kit ME10-1KT) and LC/MS-grade acetonitrile (CAS 75-05-8) were from Sigma (St. Louis, MO). LC/MS grade water was from JT Baker. N-methyl-N-(trimethylsilyl)-trifluoroacetamide (MSFTA) containing 1% trimethylchlorosilane was from Restek (Bellafonte, PA).

### 2.2 Culturing

*M. vaginatus* PCC 9802 was grown at 22 ºC in minimal Jaworski's media with a 12-hour light, 12-hour dark cycle. Light was provided by a 6500 K, 2000 lumen fluorescent source. Light flux measured at the level of the petri dishes was approximately 4.5-10 µmol photons $m^{-2}$ $s^{-1}$. A minimum of 3 replicates grown at identical conditions were used for each experiment. To determine the biopolymer flux rates, 3 samples were taken at the start and end of 12 hours growing at a given condition.

### 2.3 Respiration Measurement

      Carbon dioxide flux was determined from three biological replicates in both light and dark conditions with a Micro-Oxymax Respirometer (Columbus Instruments) in sealed 100 mL mason jars. The system monitored carbon dioxide accumulation across all samples over a 48-hour period with a time resolution of approximately 48 minutes per measurement. These values are scaled to µmol/gram biomass.

### 2.4 Biopolymer Extraction

      Immediately following respiration measurements, triplicate aliquots of *M. vaginatus* cultures (1 mL) from both light and dark conditions were removed and transferred to a pre-weighed 1.5 mL tube. Cells were pelleted by centrifuging at 3000 rcf for 3 min, resuspended in 1 mL methanol and homogenized with sterile metal ball bearings using a Mini-Beadbeater (BioSpec Products). Samples were centrifuged at 3000 rcf for 3 min and supernatant discarded. Homogenization and centrifugation were repeated for a total of three times. The remaining biomass was dried in a Savant SpeedVac SPD111V and dry weight was measured. To allow efficient hydrolysis of polyphosphate, glycogen, and PHB, 500 µL of 2M HCl was added and the mixture heated at 95°C for 1 h while shaking (1400 rpm) (Dephilippis et al., 1992;Eixler et al., 2005;Ernst and Boger, 1985). Samples were then dried and resuspended in 1 mL methanol by vortexing and bath sonication (VWR Symphony). Samples were cleared of any remaining cells by centrifuging at 3000 rcf for 3 min and filtered through 0.22 µm centrifugal filters (Nanosep MF, Pall Corporation, Port Washington, NY).



## 2.5 GC/MS Measurement of Glucose and Phosphate

An aliquot of the hydrolyzed biopolymers (10% of the original biomass) was dried for GC/MS analysis of glucose and phosphate. To each dried sample, 10 µL of 40 mg/mL methoxyamine hydrochloride in pyridine was added. The mixture was shaken for 1.5 h at 30°C and 1400 rpm. FAME standards (1 µL) (Swenson et al., 2015) in 90 µL MSTFA were added to each sample and standard. The mixture was shaken at 37°C for 30 minutes and 1400 rpm. The contents were then transferred to glass vials with micro-inserts and submitted to GC-MS analysis.

GC/MS data were acquired on an Agilent 7890 gas chromatograph (Agilent Technologies, Santa Clara, CA) and an Agilent 5977 single quadrupole as described in Swenson et al, 2015. In summary, derivatized samples were injected (0.2-0.5 µL) using a Gerstel automatic liner exchange MPS system (Gerstel, Muehlheim, Germany) into a Gerstel Cooled Injection System (CIS4) operated in splitless mode. Analytes were separated using a Rxi-5Sil MS capillary column (Restek, Bellefonte, PA) with a 0.25 mm ID Integra Guard under an initial oven temperature of 60°C, held for 1 min then ramped at 10°C/min to 310°C with a total runtime of 36 min. Glucose was detected as two peaks at 17.5 and 17.7 min and phosphate at 10.0 min. Glucose and phosphate were quantified based on a 5-point standard curve (in triplicate) from 0.5-10 µg. All measurements fell within the linear range. Calculated concentrations were then normalized to original dried biomass.

## 2.6 LC/MS Measurement of Crotonic Acid

Crotonic acid was measured by LC-MS/MS on an Agilent 1290 UHPLC (Agilent Technologies, Santa Clara, CA) coupled to a Q-Exactive mass spectrometer (Thermo Scientific, San Jose, CA). A sample volume of 2 µL was injected onto a Kinetix C18-XB column (150mm x 2.6µ x 100Å) (Phenomenex, Torrance, CA), and eluted over a linear gradient of water + 0.1% formic acid (FA) (A) to acetonitrile + 0.1% FA (B), with an initial 1 min hold at 100% A, a 7-min gradient to 100% B, followed by a 1.5 min hold at 100% B, and a 2-min re-equilibration at 100% A. Crotonic acid eluted at 2.2 min, and was detected in positive mode. Crotonic acid was quantified based on a 5-point standard curve (in triplicate) using the parent ion (87.0447 m/z in positive mode), and all sample measurements fell within the linear range (10 ng/mL to 100 µg/mL). Crotonic acid concentrations were normalized to original dry biomass weights.

## 2.7 Metabolic Network Reconstruction and Refinement

The genome of *M. vaginatus* PCC 9802 was first sequenced by the US Department of Energy Joint Genome Institute. The draft reconstruction of this genome was obtained using the Model SEED, a pipeline for generating metabolic models from genome data (Henry et al., 2010). The genome was automatically annotated using RAST (Overbeek et al., 2014). A reaction network complete with gene-protein-reaction relationships, predicted Gibb's free energy of reaction values, and an organism specific biomass reaction including non-universal cofactors such as wall components and lipids was generated. This



network was also converted into its mathematical form, allowing quantitative systems biology analysis via flux balance analysis (FBA).

Automatic annotations were further refined through manual annotation. Extensive manual refinement was performed to examine RAST predicted functions, GPR relations, and fill gaps in metabolic pathways. Detailed manual
annotation reports can be seen in supplementary information. This was performed using literature review, flux balance analysis, and the databases, KEGG and MetaCyc (Kanehisa and Goto, 2000;Karp et al., 2002). Flux balance analysis was run using the COBRA Toolbox (Becker et al., 2007). The refinement process is illustrated in Figure 1.

*M. vaginatus'* biochemical capabilities are relatively unstudied in literature to date, hence, strong sequence homology to related bacterial enzymes and to previous literature findings, were used as criteria in assigning functionalities,
gene-protein-reaction relationships, and reaction directionalities. Although photosynthetic genes were identified via automatic annotation, photosynthetic reactions were not present in the generated mathematical model. To reconcile this, and given the fact that only a few models exist for cyanobacterial oxygenic photosynthesis, the model of photosynthesis developed for the cyanobacterium *Cyanothece* sp. ATCC 51142 (iCce806) was adopted (Vu et al., 2012). In addition, to enable the analysis of biopolymer fluxes, reactions for the synthesis and metabolism of biopolymers polyphosphate,
glycogen and PHB were added.

Constraint based flux balance analysis is a common approach that abstracts the metabolic capabilities of an organism as a set of linear equations, then uses constraint based linear optimization to predict the pathway usage and flux through a network (Orth et al., 2010). In this study, a pseudo-steady state approximation Eq. (1) is used, where the accumulation of each metabolite is zero over time. $S$ is the stoichiometric matrix of equations, and $v$ is the vector of flux
values. The flux is constrained by upper and lower bound vectors, $LB$ and $UB$, in Eq. (2).

$$S \cdot v = 0 \,, \tag{1}$$

$$LB \leq v \leq UB, \tag{2}$$

Model metabolites were separated into the extracellular and cytosol (inner) domains. Sink reactions simulated transport between the environment and extracellular domains, while exchange reactions mediated transport between
extracellular and cytosol domains.

Despite initial manual curation efforts, the model was initially unable to simulate photoautotrophic growth using a minimal media using the biomass reaction from ModelSEED. A minimal media is defined as only containing essential trace minerals, oxygen, carbon dioxide, light, sulphate, and nitrate/ammonium. Missing precursors, mainly, thiamine-phosphate, L-cysteine, and spermidine, were found to prevent growth. Using the databases, KEGG and MetaCyc, possible reactions that
allow *M. vaginatus* to produce biomass were identified in a process known as "gap-filling." Once found, these functions were cross-checked to determine if they were predicted with lower homology in *M. vaginatus'* genome, and common in related cyanobacteria. If these criteria were not fulfilled, but the function was absolutely required, the reaction was still incorporated in the model with a low confidence rating.



**2.8 Modelling Biopolymer Accumulation and Utilization**

Modelling biopolymer depletion was accomplished by treating biopolymer subunits as bounded external resources. In other words, instead of the typical accumulation reaction, "pool" reactions, Eq. (4) and Eq. (5), were created for each biopolymer, where $A$ is a side reactant, $B$ is a side product, $X$ is a polymer subunit, and $X_n$ is a polymer with length $n$. The sink reaction, Eq. (5), then allows the model to undergo steady state simulation. The fluxes through the sink reactions from FBA simulation represent rates of accumulation or depletion.

$$X + \rightarrow X_n + B, \tag{4}$$

$$X_n + X + A \leftarrow\rightarrow X_{n+1} + B, \tag{3}$$

$$X_n \rightarrow nothing, \tag{5}$$

        To predict respiration in the dark, the light flux rate was set to 0 and biopolymer flux rates were constrained by the experimentally determined values. To set the flux of an individual reaction, Eq. (6) was used, where $UB_i$ and $LB_i$ are the

upper and lower constraints of reaction $i$, and $v_i^{exp}$ is the measured flux.

$$UB_i = LB_i = v_i^{exp}, \tag{6}$$

        Modelling biopolymer accumulation under light was accomplished by including biopolymer requirements within the biomass objective function, according to their stasis levels as measured during experiments. Biopolymer accumulation rates can then be calculated using the simulated biomass accumulation rate.

180        The ability of the constructed network to quantitatively predict metabolic activity was assessed by comparing the calculated rates of respiration in the dark and $CO_2$ uptake and biopolymer accumulation in the light, with the experimentally determined values.

        To address the issue of biological variation across samples and their impact on model predictions, sensitivity analysis was performed on the flux constraints. Biopolymer constraints were varied within two standard deviations of the

average measured value, light flux levels were varied within $\pm$ 1 µmol photons m$^{-2}$ s$^{-1}$, and biomass levels were varied within $\pm$ 2 mg. A total of 30,619 different simulations were performed in this analysis.

**3 Results & Discussion**

**3.1 Metabolic Network Reconstruction**

The resulting reconstructed genome scale model possesses 858 genes, 1153 reactions, and 1078 metabolites. A total of 146

reactions were added from manual curation, gap analysis, and for biopolymer simulations of polyphosphate, glycogen, and PHB. For a comparison of the automated and curated annotation see Table 1. Added biopolymer reactions are shown in Table 2. To view the model in Excel and Matlab structure formats see the supplementary information.




### 3.2 Experimentally Determined Fluxes During Photosynthesis and Respiration

In both light and dark conditions, carbon dioxide profiles varied linearly with time, supporting the assumption that

*M. vaginatus'* metabolism behaves like a quasi-steady-state system over 12-hour time cycles. It also indicates that using experimental data to validate and constrain a steady state flux balance model is reasonable in a 12-hour time scale.

With a light flux of approximately 141 µmol/mg/h, *M. vaginatus* accumulated carbon dioxide at approximately 19.6 ± 3.7 µmol/g biomass/hour.  Without light, it released carbon dioxide at 17.3 ± 3.2 µmol/g biomass/hour. The biopolymer flux rates in the dark and under a light flux of 4670 µmol/mg/h are reported in Table 3.

Interestingly, the rate of glycogen depletion in the dark is approximately a quarter of the rate of accumulation in the light.  For these conditions, *M. vaginatus* has a conservative metabolism, accumulating much more biopolymer in a day than is necessary for the following night.  We can speculate that biopolymer reserves may be accumulated such that sufficient quantities may drive physiological responses such as cell division, making it akin to the yeast metabolic cycling observed for low glucose conditions (Yin et al., 2003). Since this photoautotroph is limited by nutrients other than carbon, they are likely

routing flux from intracellular biopolymers into EPS formation, consistent with our earlier results showing export of a diversity of oligosaccharides (Baran et al., 2013).

Gene expression studies showed increased expression of hydrolytic enzymes (e.g. glycogen debranching enzyme) during dry-down which may provide a mechanism to rapidly produce compatible solutes (Baran et al., 2017). These are presumably excreted during wet-up resulting in an additional metabolic cost for wet-up that must be recouped through

biopolymer accumulation. Thus, the observed high flux to glycogen may provide multiple physiological adaptations that are critical to survival in arid climates.

Polyphosphate possesses an uptake rate like that of the glycogen, but is only depleted 1% as fast in the dark, indicating that this resource is conserved quite well. Polyphosphate is likely an important biopolymer for M. vaginatus across many different stressed conditions, and may be used in other ways than as an energy source, which agrees with gene

expression studies (Rajeev et al., 2013).

PHB accumulates at roughly 0.4 % of the rate that glycogen accumulates, indicating that it is not the primary polymer used for storage. Interestingly, its depletion rate is higher than its accumulation rate. It may be used as a secondary storage polymer or for other metabolic activities that do not require a consistent energy source, such as replication. The study of PHB over multiple light/dark and dry/wet cycles may be crucial to understanding the role that PHB plays.

Although each bacterial sample was taken from the same culture, variation in metabolic activity existed regardless of dry biomass normalization.  This may be attributed to differing amounts of polysaccharide sheaths surrounding active cells (Bertocchi et al., 1990). The large amounts of exopolysaccharides in *M. vaginatus (Hokputsa et al., 2003)*, which are considered distinct in chemical form and function from glycogen, may also interfere with GC-MS quantification of storage glucose.  A complete study of EPS composition in *M. vaginatus* has yet to be completed in the authors' existing knowledge.

Therefore, we have limited our analysis to using these measurements as the upper and lower bounds on metabolic fluxes.



Improvements in methods for quantifying glucose levels from glycogen without EPS bias will be important for further model refinement as would be extension of the model to include EPS.

### 3.3 Simulation Results and Experimental Validation

A comparison of simulation and experimental $CO_2$ accumulation rates can be seen in Figure 2. In the dark, the predicted average carbon dioxide flux was 11.9 μmol/day/g, close to the measured value of 17.3 ± 3.2 μmol/day/g. In the light, the predicted average carbon dioxide uptake flux using the measured light flux as a constraint was -33.4 μmol/g biomass/hour, compared to the experimental value of -19.6 ± 3.7. Sensitivity analysis showed that the variation in biopolymer flux measurements, illumination, and biomass could account for this variation.

Accumulation rates of polyphosphate and PHB fall within the range predicted by simulation. The average simulated glycogen accumulation is lower than the experimentally measured glycogen accumulation rate, though sensitivity analysis may account for this. Another possible reason for this lower estimate could be that *M. vaginatus* biomass requirements are less than the requirements given by ModelSEED. To verify the biomass composition a more thorough investigation may be conducted.

From the modelled fluxes, we estimate that under constant wetted conditions, *M. vaginatus* routes 2% of its carbon uptake to the generation of storage glycogen and PHB, while the rest may be attributed to cell maintenance, biomass growth, or other unknown storage molecules. Although this value is expected to change depending on growth conditions, it provides a basis for future studies. Further analysis of the network fluxes may prove useful in investigating potentially important pathways in *M. vaginatus*' metabolism. For such studies, flux information from light and dark experiments across all
reactions have been included in the supplementary section.

Compared to the metabolism of *Synechocystis* sp. PCC 6803 modelled by Knoop *et al*, $CO_2$ uptake and glycogen accumulation per photon of light are 2.5 and 5.7 times faster respectively. Glycogen depletion in the dark is only 36% faster. The accelerated accumulation of carbon indicates that *M. vaginatus* may have a more efficient pathway for converting $CO_2$ to biopolymer storage, which a possible reason for its role as a pioneer organism in BSCs.


### 3.4 Survivability during Pulsed Activity Events

BSCs are vulnerable to alteration in wetting events (Reed et al., 2012). This coupled with predictions of alterations in rainfall and increasing aridity in biocrust habitats (Maestre et al., 2015a;Maestre et al., 2015b) makes it important to develop tools to predict survival under changing rainfall and light intensity. Towards this goal, we can use the results from
our model as a crude estimate of the 'health' of *M. vaginatus* based on biopolymer levels. Certainly, this is overly simplistic and there are many other factors that impact 'health'. However, given the central role of biopolymers in surviving in the dark and producing compatible solutes, their levels are expected to be critical to survival. Because glycogen, the primary carbon



biopolymer, is consumed at about 25% of the accumulation rate, if wetting events in the dark are more than four times as long as wetting events in the light, *M. vaginatus'* health would begin to deteriorate. For lower light intensities than the ones used here, the rate of glycogen accumulation would decrease, reducing *M. vaginatus* resistance to persistent dark wetting events.

The model constructed here may be applied to more complicated, dynamic events, such as those with varied intensities of light and wetting time. To illustrate how this might be achieved, we simulated glycogen depletion over 100 days for various wetting frequencies in the light and dark, seen in Figure 3, where for each day there is either a 12-hour wetting event. When the ratio of light:dark events of 1:3, glycogen storage is maintained, whereas at 1:1 accumulation is rapid. At 1:5 glycogen stores become depleted after 94 days. This simulation may be improved in future studies by considering dynamic responses to light variation as well as the drying and re-wetting of soil.

**Conclusions**

We have generated the first reconstruction of the *M. vaginatus* metabolic network through careful manual annotation and inclusion of biopolymer reactions known to be important to cyanobacterial physiology. We use the model within a conceptual framework of "costs" and "benefits" associated with specific environmental changes. More precisely, we observed that dark reactions have a 'cost' which is recouped during light reactions. Using these experimental constraints on biopolymer and light fluxes, the model generates $CO_2$ and biopolymer fluxes that match experimental results. Using this model and laboratory experiments we found that, in light, 2% of carbon uptake is routed to storage polymers via photosynthesis. In the dark, glycogen functions as the primary carbon source. Using depletion and accumulation rates, we then predicted the gradual depletion of *M. vaginatus'* biopolymer reserves if dark wetting events were four times as long as light wetting events. This work lays the foundation for additional research on this important desert microbe and its microbial community. Since *M. vaginatus* is a keystone organism in many biocrusts, it's loss would have a major impact on the community. Yet, it is important to emphasize that these laboratory experiments are a vast extrapolation from native conditions and future work will focus on extending these models to more realistic environmental conditions. Additional steady-state experiments using different growth conditions found in desert environments—varying lighting intensities, levels of moisture, soil compositions, and other microbial interactions—have the potential to refine the model and identify key metabolic states related to resuscitation and dormancy to more accurately model environmental responses.

**Abbreviations**

BSC             Biological Soil Crust

EPS             Exopolysaccharides

FBA             Flux Balance Analysis



GC/MS  Gas Chromatography Mass Spectrometry

Acc.              Accumulation

Resp.             Respiration

*Chemicals*

ADP              Adenosine diphosphate

ATP              Adenosine triphosphate

FAME             Fatty Acid Methyl Ester

MeOX             Methoxyamine hydrochloride

MSFTA N-methyl-N-(trimethylsilyl)-trifluoroacetamide

PHB              β-polyhydroxybutyrate

Gly              Glycogen

PP               Polyphosphate

*Notation*

$A$              generic side reactant

$B$              generic side product

$LB_i$           Flux lower bound of reaction $i$

**$LB$**          Vector of flux lower bound values

$P$              phosphate level

$S$              Stoichiometry matrix of reaction equations

$UB_i$           Flux upper bound of reaction $i$

$UB$             Vector of flux upper bound values

$v_i^{exp}$      Experimentally determined flux value of reaction $i$

$v$              Vector of flux values

$X$              Polymer precursor

$X_n$            Polymer subunit

**Competing interests**

The authors declare that they have no conflict of interest.



## Acknowledgements

We acknowledge the contribution of Seth Axen, Rahul Basu, Kriti Sondhi, and David Soendjojo with their assistance annotating the *M. vaginatus* genome. This work was supported in part by previous breakthroughs obtained through the Laboratory Directed Research and Development Program of Lawrence Berkeley National Laboratory supported by the US

Department of Energy Office of Science and through the US Department of Energy Office of Science, Office of Biological and Environmental Research Early Career Program (award to T.R.N.) both under contract number and DE-AC02-05CH11231.

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

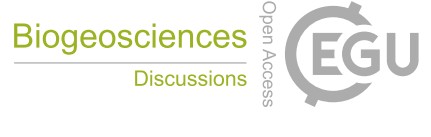

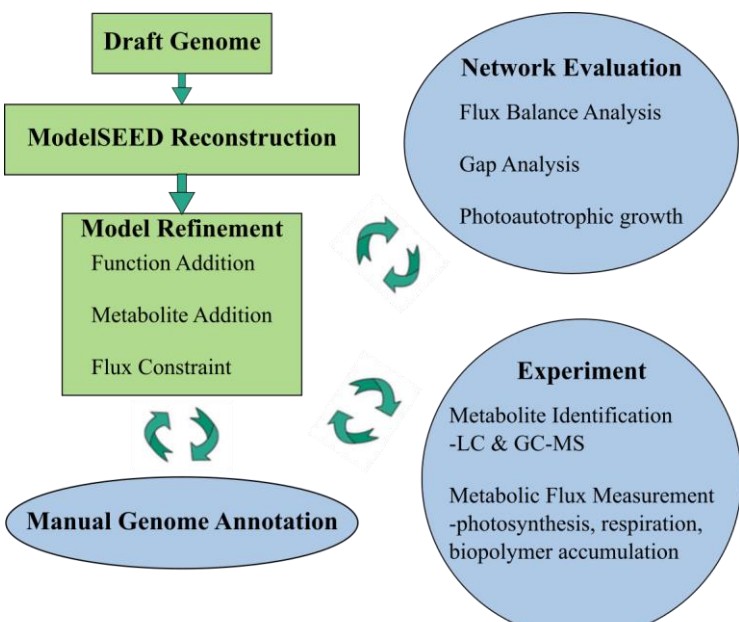

**Figure 1: Metabolic reconstruction process diagram.**





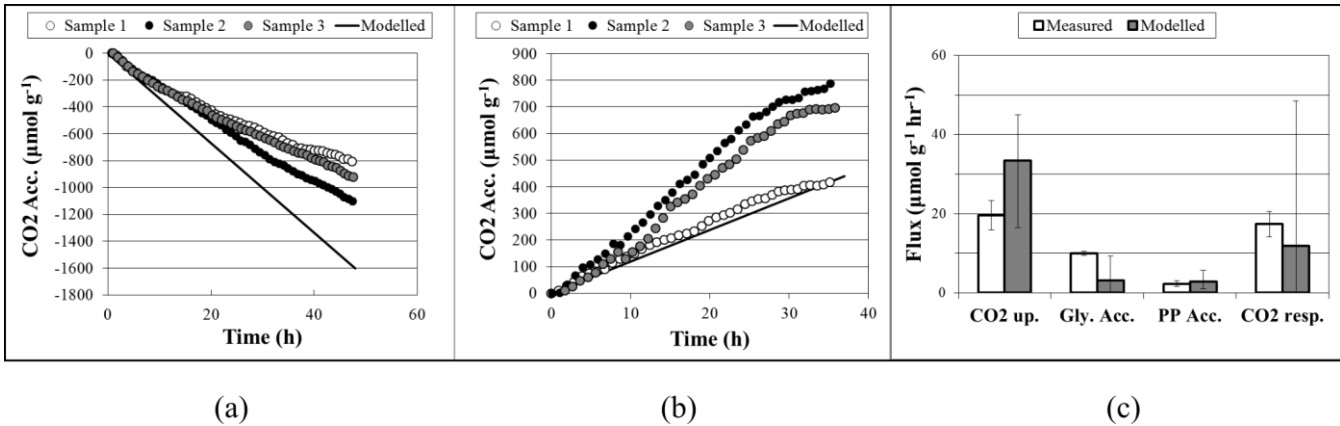

(a)  (b)  (c)

**Figure 2: Comparisons of experimental and modelled $CO_2$ accumulation (a) in the light, where a negative value indicates uptake, and (b) in the dark, where a positive value indicates respiration. Biopolymer and $CO_2$ flux rates are compared in (c), where error bars on modelled flux rates are the upper and lower bounds determined through sensitivity analysis; error bars on measured flux rates are standard deviations.**






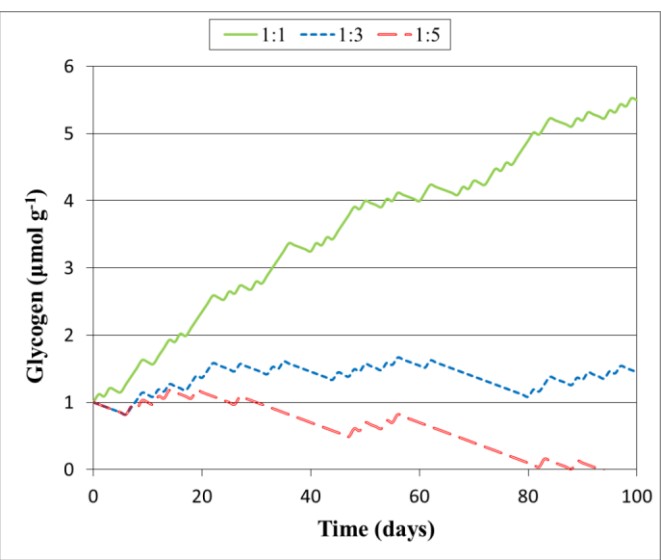

**Figure 3: Predicted glycogen depletion over 1 month under three climate scenarios, where the ratio of light:dark wetting events are 1:1, 1:3, and 1:5.**






**Table 1: Comparison of curation and automated annotation.**

| Major manually curated pathways | Predicted in curation | Predicted in RAST annotation |
|---|---|---|
| All amino acid biosynthetic pathways | x | x |
| Ammonium Assimilation | x | x |
| Bifidobacterium Shunt | x | x |
| Heterolactic Acid Fermentation | | |
| Homolactic Acid Fermentation | x | x |
| Mixed Acid Fermentation | | |
| Nucleoside triphosphate biosynthetic pathways | x | x |
| Photosynthetic light reactions | x | |
| Calvin Cycle | x | x |
| Nitrate Assimilation | x | |
| Nitrogen Fixation | | |
| Glycogen Biosynthesis | x | |
| Glycolysis | x | x |
| Hydrogen Production | | |
| Pentose Phosphate Cycle | x | x |
| Sulfur and Sulfate Reduction | x | x |
| TCA Cycle | x | x |
| Peptidoglycan Biosynthesis | x | x |
| β-polyhydroxybutyrate synthesis | x | |
| Cyanophycin synthesis | x | |
| Polyphosphate synthesis | x | |



**Table 2: Modelled Biopolymer Reactions.**

| Reaction | Description |
|---|---|
| **ATP ←→ADP + Polyphosphate** | Polyphosphate synthesis/degradation |
| **Polyphosphate ←→ Nothing** | Polyphosphate sink |
| **Glucose-1-phosphate ← Phosphate + H(+) + Glycogen** | Glycogen degradation |
| **ADP-Glucose → Glycogen + ADP** | Glycogen synthesis |
| **Glycogen ←→Nothing** | Glycogen Sink |
| **(R)-3-Hydroxybutanoyl-CoA ←→ CoA+PHB** | PHB synthesis/degradation |
| **PHB←→Nothing** | PHB sink |




**Table 3: Experimental and modelled flux values over light and dark conditions. Constraint fluxes are noted with a "*". Negative and positive CO₂ fluxes represent uptake and respiration respectively, while negative and positive biopolymer flux rates represent depletion and accumulation respectively.**

| | Measured | | | Modeled | | |
|---|---|---|---|---|---|---|
| | Flux ($\mu mol\ g^{-1}\ h^{-1}$) | LB | UB | Flux ($\mu mol\ g^{-1}\ h^{-1}$) | LB | UB |
| **Light (1)** | | | | | | |
| Light* | 141 | | | 141 | 93 | 211 |
| $CO_2$ | -19.6 | -23.3 | -15.9 | -33.4 | -50.3 | -21.8 |
| **Light (2)** | | | | | | |
| Light* | 4670 | | | 4670 | 3551 | 6290 |
| Glycogen | 9.99 | 9.47 | 10.51 | 3.07 | 0.00 | 9.30 |
| PHB | 0.0366 | 0.0050 | 0.0683 | 0.121 | 0.044 | 0.250 |
| Polyphosphate | 2.28 | 1.57 | 2.99 | 2.82 | 1.00 | 5.70 |
| **Dark** | | | | | | |
| Light* | 0.00 | | | 0.00 | | |
| Glycogen* | -2.49 | -6.08 | 1.09 | -2.49 | -9.66 | 4.67 |
| PHB* | -0.14 | -0.17 | -0.11 | -0.136 | -0.20 | -0.07 |
| Polyphosphate* | -0.02 | -1.27 | 1.22 | -0.02 | -2.51 | 2.46 |
| $CO_2$ | 17.3 | 14.1 | 20.6 | 11.9 | 0 | 48.5 |
