# Peer review of "Flux balance modelling to predict bacterial survival during pulsed activity events"

_Biogeosciences, 2017_

## Referee Comment (RC1) · Anonymous Referee #1 · 31 Oct 2017

The authors present a thoroughly documented manual reconstruction of Microcoleus vaginatus, a terrestrial cyanobacterium adapted to arid environments. The reconstruction is experimentally validated by comparing predicted $CO_2$ and storage polymer production & consumption in the light and dark to measured values. The metabolic model is then exploited to predict glycogen concentration in M. vaginatus under different climate scenarios. This is solid work and I have only a few comments.

What is the number and percentage of genes with unknown functions in M. vaginatus?

How well do the predicted fluxes during light and dark agree with previous results on gene expression in M. vaginatus-dominated BSCs?

A delicate point in FBA is the definition of the objective function. The Materials &

[Figure]

Methods part does not state clearly which function(s) is (are) optimised - is it biomass formation or flux through polymer-forming reactions or both? In case of biomass, how was the biomass formation reaction obtained? How well does the predicted light and dark growth rates (corresponding to the flux through the biomass formation reaction) agree with measured growth rates?

Is it known whether the ratio of day/night wetting events or day length influences the distribution of M. vaginatus in deserts and arid environments?

The authors should clarify whether or not upper and lower flux bounds are constrained by measurements when predicting flux through biopolymer reactions in the dark (Table 3). How well is $CO_2$ production and biopolymer consumption predicted without constraining flux boundaries with measurements?

It is curious that M. vaginatus grows so slowly (weeks) and appears to miss vital reactions. Is it possible that it relies on near-obligate mutualistic partners in nature?

Please also share the metabolic model as an SBML file. Not everyone has access to Matlab.

l. 67: a terrestrial cyanobacteria -> a terrestrial cyanobacterium l. 249: which a possible reason -> which is a possible reason l. 264: there is either a 12-hour wetting event -> or?

---

## Author Comment (AC1) · 18 Nov 2017

Dear Referee,

Thank you for your comments.

Out of 5265 putative genes, there are 1990 genes (37.8%) with unknown functions from initial automated annotation. The 858 genes currently in the model are picked from initial assignments via additional annotation via the ModelSEED pipeline and manual efforts.

The predicted fluxes compare well to the ones we measured in Rajeev et al (2013). In the light BSCs had a maximum uptake of approximately 12 – 20 $\mu$mol $CO_2$ gˆ-1 hrˆ-1, compared to the predicted average of 11.9 $\mu$mol $CO_2$ gˆ-1 hrˆ-1. In the dark BSCs

had a maximum production of approximately $12 - 20$ $\mu$mol CO2 gˆ-1 hrˆ-1, compared to the predicted average of 33.4 $\mu$mol CO2 gˆ-1 hrˆ-1. Although there is only a slight difference, the comparison is difficult to draw conclusions from because gene expression studies used BSC samples in field-like conditions, whereas our studies used M. vaginatus grown in isolation, constantly wetted in minimal media. In addition, because biomass was not used to normalize flux measurements in gene expression studies, an unknown biomass density needs to be assumed.

The biomass objective function is optimized. The function was initially obtained through ModelSEED, and was modified to include biopolymers in the light as a biomass requirement. The rate of total biomass accumulation was not measured, so that comparison could not be made. We do observe that the predicted growth rates scale with increasing light intensity and that in the absence of any light in the model the predicted growth rate is 0. Both of these observations are constant with the known growth patterns of M. vaginatus.

There are no experimental studies to our knowledge that have specifically investigated the ratio of day/night wetting events. The reason for such a simulation is more to demonstrate how this model could be used given the information we have determined–carbon fluxes at fixed time intervals in the light and dark. This was inspired by the previous studies of Belnap et al (2003) and Reed et al (2012), who showed that wetting frequency and season affect the carbon balance and production of pigments for radiation-protection. A follow-up study could explore this day/night ration and wetting frequency across seasons.

Upper and lower flux bounds were constrained by the average flux measurement in the dark. The description of Table 3 will be modified to include this clarification. Using an unconstrained flux boundary for resources (represented as positive and negative infinity for the upper and lower bound) would result in an unconstrained/infinite CO2 flux. The purpose of constraining the FBA problem is to reduce the degrees of freedom to solve for a desired unknown flux.

M. vaginatus certainly has mutualistic partners in nature (and in culture) that likely facilitate more rapid growth, but they are not essential for survival because M. vaginatus may be grown axenic. The naturally slow growth is also thought to be an evolved mechanism that allows survival in harsh arid environments. The "missing" of vital reactions is more a result of annotation, where key genes in M. vaginatus lack known homologs for functional assignment.

The suggested corrections will be added:

An SBML file (see attached) will be added to the supplementary information. Lines 67 and 249 will be corrected. Line 264: where for each day there is either a light or dark 12-hour wetting event.

Please also note the supplement to this comment:
https://www.biogeosciences-discuss.net/bg-2017-403/bg-2017-403-AC1-supplement.zip

―――――――――――――――

---

## Referee Comment (RC2) · Anonymous Referee #1 · 22 Nov 2017

The authors have responded adequately to all my questions and comments. I have one last remark, though: The SBML file generates a couple of warnings when loaded with R package sybilSBML. More specifically, two reactions appear to miss an identifier (no_id) and another reaction with identifier rxn00062 occurs twice. In addition, the number of genes and metabolites displayed in R and Matlab (with iNJ1153.mat) is not the same. I hope the authors can look into these issues before publication.

---

## Referee Comment (RC3) · Anonymous Referee #1 · 5 Dec 2017

The authors have addressed all my concerns.

---

## Author Comment (AC2) · 5 Dec 2017

Dear Referee,

Thank you for your response and apologies for the issues regarding the SBML file. Please find attached a version that should run without errors in R. Modifications were minor. The number of unique genes does not reflect the number of total genes in the model due to redundancy in annotation, where more than one gene may be associated with the same function.

Please also note the supplement to this comment:

[Figure]

https://www.biogeosciences-discuss.net/bg-2017-403/bg-2017-403-AC2-supplement.zip

---

## Referee Comment (RC4) · Anonymous Referee #2 · 30 Jan 2018

Flux balance modeling to predict bacterial survival during pulsed activity events by Jose et al.

This is a very interesting study where genome information is combined with detailed measurements of storage products to reach the quantitative conclusion about the cost of activity in the dark and the light.

Love the paper, well written, full of new information (for me). Not being familiar with the details of genome and flux balance approaches, my questions are mostly related to clarification.

1) in the metabolic model, I assume all growth functions, such as RNA, and protein production are included? Does the model assume that these other functions, such as

protein production etc, happen at similar rates during the day as during the night?

2) The organisms were cultured at 4.5-10 umol m-2 s-1 (L83). How does this compare to the desert light levels, and how would an increase in light level affect the conclusion of this paper regarding wet/dry cycles during day/night? Is the light level PAR or total radiation? Related topic, in the results and discussion (L 198, 199; Table 3) two light levels are defined. It was not clear when reading the paper what this meant, whether this was caused by a change in biomass production or changed light levels. Please clarify.

3) L 116: please clarify why crotonic acid was determined. As far as I know, it is not mentioned in the results and discussion but seems to be the breakdown product of polyhydroxybutyrate. (?)

4) As a clarification, please explain why biopolymer reactions are included in the model, but not found in the genome. Similar for the other processes.

5) L 150: how were LB and UB determined. On the one hand they seem to be the product of the model (Table

3) but at the same time constrain the model (L 150).

6) Table 1: what is the relevance of the reactions mentioned such as hydrogen production in the table, but neither curated not found in the genome.

7) L 170-173 and Table 2: I am not familiar with flux balance calculations, so the phrase PHB ßà nothing is confusing? Please add explanation in one additional sentence.

8) Table 3 and Fig. 2c seem to have some overlap.

9) L 214 Âň please elaborate how polyP can be used in other ways than an energy source

10) L 218: change consistent to constant or words similar to that.

L 52: diel does not need to be capitalized

L 63: the use of the phrases dark and light reactions are confusing: they have very specific meaning in the study of photosynthesis, but I don't think that is what is meant here. Please Âň replace with something like metabolism in the dark versus in the light.

L71 Âň complex sentence that can be simplified.

L 94 Âň add rcf to the list of abbreviations, and add units

L 134 Âň what are GPR relations

L 135: comma after databases can be removed

L 168: why the word "side" with reactant and product?

L 194 Âň change profiles to concentrations

L 246: Add year after reference (Knoop)

L 247: is the efficiency measured at the same light level?

---

## Author Comment (AC3) · 16 Feb 2018

1) In the metabolic model, I assume all growth functions, such as RNA, and protein production are included? Does the model assume that these other functions, such as protein production etc, happen at similar rates during the day as during the night?

Yes, production of RNA and protein are also included. The rate of production is related to the intensity of light, so that these rates are different during the day and night.

2) The organisms were cultured at 4.5-10 umol m-2 s-1 (L83). How does this compare to the desert light levels, and how would an increase in light level affect the conclusion of this paper regarding wet/dry cycles during day/night? Is the light level PAR or total radiation?

While the radiation incident on the crust surface can be orders of magnitude higher that the one we used, it is subject to intense multiple scattering losses, so that only 1 percent of incident radiation remains a couple of millimeters down into the soil. *M. vaginatus* makes a living within this steep light gradient, usually in the subsurface, coming up to the surface only when light intensity is very moderate, (morning, overcast conditions) and it has a "shade plant" phenotype (low photosynthesis saturation intensity, heavy complement of light harvesting pigments and no sunscreen pigments). Isolate *M. vaginatus* does not grow in liquid media under desert light levels that we used previously to mimic more natural conditions in the lab with intact biocrusts (~600 umol m-2 s-1; DOI: 10.1038/ismej.2013.83).
We conducted our experiments close to the growth optimum of *M. vaginatus* in liquid media. The light level given is PAR.

This text has been added to the manuscript in the methods section (section 2.2)

Related topic, in the results and discussion (L 198, 199; Table 3) two light levels are defined. It was not clear when reading the paper what this meant, whether this was caused by a change in biomass production or changed light levels. Please clarify.

Thanks for pointing this out. The manuscript has been changed to clarify this:
L83: Because different culture containers were used between respiration experiments and biopolymer experiments, they possessed different photon fluxes (measured in $\mu mol\ g^{-1}\ h^{-1}$), which is reflected in Table 3. This is accounted for in simulations.

3) L 116: please clarify why crotonic acid was determined. As far as I know, it is not mentioned in the results and discussion but seems to be the breakdown product of polyhydroxybutyrate. (?)

The manuscript has been changed to clarify this:

Section 2.6 Title: "Quantification of PHB via LC/MS Measurement of Crotonic Acid"
L117: "Polyhydroxybutyrate breaks down to form crotonic acid in strongly acidic environments. The quantity of crotonic acid formed is used to calculate PHB quantity."

4) As a clarification, please explain why biopolymer reactions are included in the model, but not found in the genome. Similar for the other processes.

These clarifications have been added to the manuscript:

L133: "Automatic annotations were further refined through manual annotation. This was necessary because automated modelling databases occasionally do not contain strong homologs for a certain function, and thus fail to assign it to the genome via simple homology search algorithms."

L144: "…reactions for the synthesis and metabolism of biopolymers polyphosphate, glycogen and PHB, which were not given within the automated annotation, were added after being found in the genome through extensive manual annotation."

L165: "Automated model generation typically does not assume that biopolymers act as resources that may accumulate or deplete."

5) L 150: how were LB and UB determined. On the one hand they seem to be the product of the model (Table 3) but at the same time constrain the model (L 150).

In Table 3 measured UB and LB refer to the experimental bounds, i.e. the standard deviation. Modelled UB and LB refer to the upper and lower values obtained from sensitivity analysis. In modelled constraint reactions for light, UB and LB represent the assumed deviation that is input into sensitivity analysis.

The reaction flux UB and LB used as inputs in the model are fixed at the experimentally measured values (see L176, eqn 6). In sensitivity analysis, this is varied over the estimated deviation (See L182-186).

Due to this confusion, the LB and UB in Table 3 has been amended as seen below.

**"Table 1: Experimental and modelled flux values over light and dark conditions. Constraint fluxes are noted with a "*". Negative and positive $CO_2$ fluxes represent uptake and respiration respectively, while negative and positive biopolymer flux rates represent depletion and accumulation respectively. Measured "-" and "+" refer to the standard deviation. Modeled "-" and "+" refer to the upper and lower values obtained from sensitivity analysis. In modeled constraint reactions for light, "-" and "+" represent the assumed deviation that is input into sensitivity analysis."**

| Measured | | | Modeled | | |
|---|---|---|---|---|---|
| Flux ($\mu$mol g$^{-1}$ h$^{-1}$) | - | + | Flux ($\mu$mol g$^{-1}$ h$^{-1}$) | - | + |

| | | | | | | |
|---|---|---|---|---|---|---|
| **Light (1)** | | | | | | |
| Light* | 141 | | | 141 | 93 | 211 |
| $CO_2$ | -19.6 | -23.3 | -15.9 | -33.4 | -50.3 | -21.8 |
| **Light (2)** | | | | | | |
| Light* | 4670 | | | 4670 | 3551 | 6290 |
| Glycogen | 9.99 | 9.47 | 10.51 | 3.07 | 0.00 | 9.30 |
| PHB | 0.0366 | 0.0050 | 0.0683 | 0.121 | 0.044 | 0.250 |
| Polyphosphate | 2.28 | 1.57 | 2.99 | 2.82 | 1.00 | 5.70 |
| **Dark** | | | | | | |
| Light* | 0.00 | | | 0.00 | | |
| Glycogen* | -2.49 | -6.08 | 1.09 | -2.49 | -9.66 | 4.67 |
| PHB* | -0.14 | -0.17 | -0.11 | -0.136 | -0.20 | -0.07 |
| Polyphosphate* | -0.02 | -1.27 | 1.22 | -0.02 | -2.51 | 2.46 |
| $CO_2$ | 17.3 | 14.1 | 20.6 | 11.9 | 0 | 48.5 |

6) Table 1: what is the relevance of the reactions mentioned such as hydrogen production in the table, but neither curated not found in the genome.

Fermentation and nitrogen fixation are known to be important pathways in biocrust metabolism. We annotated these to investigate if *M. vaginatus* could play a key role. Hydrogen production was annotated because hydrogen evolution was measured from crusts in previous experiments, though not reported.

7) L 170-173 and Table 2: I am not familiar with flux balance calculations, so the phrase PHB -> nothing is confusing? Please add explanation in one additional sentence.

This has been changed:

L167: "…where $A$ is a side reactant, $B$ is a side product, $X$ is a polymer subunit, and $X_n$ is a polymer with length $n$. 'Nothing" is not a physical term, but a mathematical way to describe resource accumulation in a steady state simulation."

8) Table 3 and Fig. 2c seem to have some overlap.

This is true, they are just different representations of the same data, we've decided it best to keep them because they may be useful to different types of readers. While the graph provides a visual comparison of experiment and model predictions, the tabulation of values may be helpful for those performing future simulations and studies.

9) L 214: please elaborate how polyP can be used in other ways than an energy source
This has been corrected as described below:

L213-214: "Polyphosphate is likely an important biopolymer for *M. vaginatus* across many different stressed conditions as a reservoir of phosphate for later growth, though so-called "luxury uptake" and storage when growth is halted by some other factor, and as a reservoir of energy in the form of phosphate-phosphate bonds under conditions of abundant energy generation, phosphate and a lack of conditions to use it for growth or homeostasis. This importance of polyphosphate has been identified in gene expression studies (Rajeev et al., 2013)."

10) L 218: change consistent to constant or words similar to that.

Done

L218: "…storage polymer or for other metabolic activities that do not require a constant energy source, such as replication."

L 52: diel does not need to be capitalized

Done

L52: "During the diel cycle…"

L 63: the use of the phrases dark and light reactions are confusing: they have very specific meaning in the study of photosynthesis, but I don't think that is what is meant here. Please ¡n replace with something like metabolism in the dark versus in the light.

Good point. We avoid those terms now in the manuscript to avoid confusion.

L63: "Therefore, we expect wet-up and dry-down metabolism in the dark likely have fixed biopolymer costs whereas metabolism in the light enables replenishment of biopolymer reserves."

L71: complex sentence that can be simplified.
Done

L70-71: "We interpret these results using a simple cost/benefits framework. The "cost" is biopolymer depletion in the dark, and the "benefit" is biopolymer accumulation in the light.

L 94: add rcf to the list of abbreviations, and add units
Done
Changed to "x g" (g-force). All units will be added to the abbreviations.

L 134: what are GPR relations

The Gene-Protein-Reaction relation. It is a standardized description of the link between a gene, its associated protein, and the associated reaction in genome-scale metabolic models. We will include this in abbreviations.

L 135: comma after databases can be removed
Done
L135: "…analysis, and the databases KEGG and MetaCyc…"

L 168: why the word "side" with reactant and product?

The reactants and products of interest are the biopolymer and its subunit. A and B are used to generalize other chemicals involved in the process.

L 194: change profiles to concentrations
Done
L194: "…carbon dioxide profiles concentrations varied linearly with time…"

L 246: Add year after reference (Knoop)
Done
L246: "Knoop *et al.* (2013)"

L 247: is the efficiency measured at the same light level?

No, the rates are normalized to the photon intensity in this comparison.

---

## Author Comment (AC4) · 16 Feb 2018

Thank you for your detailed questions and suggestions. We have amended the manuscript with the changes mentioned in previous author comments.

---

## Author Response (AR1)

**Flux balance modelling to predict bacterial survival during pulsed activity events**

Nicholas A. Jose[1], Rebecca Lau[1], Tami L. Swenson[1], Niels Klitgord[1], Ferran Garcia-Pichel[2], Benjamin P. Bowen[1], Richard Baran[1], Trent R. Northen[1]

[1]Environmental Genomics and Systems Biology Division, Lawrence Berkeley National Laboratory, Berkeley, California 5 94720, United States

[2]School of Life Sciences, Arizona State University, Tempe, Arizona 85287, United States

Correspondence to: Trent Northen (TRNorthen@lbl.gov)

**Reviewer Comments**

31 Oct 2017
**Reviewer:** The authors present a thoroughly documented manual reconstruction of Microcoleus vaginatus, a terrestrial cyanobacterium adapted to arid environments. The reconstruction is experimentally validated by comparing predicted CO2 and storage polymer production & consumption in the light and dark to measured values. The metabolic model is then exploited to predict glycogen concentration in M. vaginatus under different climate scenarios. This is solid work and I have only a few comments.
What is the number and percentage of genes with unknown functions in M. vaginatus?

**Authors:** Out of 5265 putative genes, there are 1990 genes (37.8%) with unknown functions from initial automated annotation. The 858 genes currently in the model are picked from initial assignments via additional annotation via the ModelSEED pipeline and manual efforts.

**Reviewer:** How well do the predicted fluxes during light and dark agree with previous results on gene expression in M. vaginatus-dominated BSCs?

**Authors:** The predicted fluxes compare well to the ones we measured in Rajeev *et al* (2013). In the light BSCs had a maximum uptake of approximately $12 - 20$ µmol $CO_2$ $g^{-1}$ $hr^{-1}$, compared to the predicted average of 11.9 µmol $CO_2$ $g^{-1}$ $hr^{-1}$. In the dark BSCs had a maximum production of approximately $12 - 20$ µmol $CO_2$ $g^{-1}$ $hr^{-1}$, compared to the predicted average of 33.4 µmol $CO_2$ $g^{-1}$ $hr^{-1}$. Although there is only a slight difference, the comparison is difficult to draw conclusions from because gene expression studies used BSC samples in field-like conditions, whereas our studies used *M. vaginatus* grown in isolation, constantly wetted in minimal media. In addition, because biomass was not used to normalize flux measurements in gene expression studies, an unknown biomass density needs to be assumed.

**Reviewer:** A delicate point in FBA is the definition of the objective function. The Materials & Methods part does not state clearly which function(s) is (are) optimised - is it biomass formation or flux through polymer-forming reactions or both? In case of biomass, how was the biomass formation reaction obtained? How well does the predicted light and dark growth rates (corresponding to the flux through the biomass formation reaction) agree with measured growth rates?

**Authors:** The biomass objective function is optimized. The function was initially obtained through ModelSEED, and was modified to include biopolymers in the light as a biomass requirement. The rate of total biomass accumulation was not measured, so that comparison could not be made. We do observe that the predicted growth rates scale with increasing light intensity and that in the

absence of any light in the model the predicted growth rate is 0. Both of these observations are constant with the known growth patterns of M. vaginatus.

**Reviewer:** Is it known whether the ratio of day/night wetting events or day length influences the distribution of M. vaginatus in deserts and arid environments?

**Authors:** There are no experimental studies to our knowledge that have specifically investigated the ratio of day/night wetting events. The reason for such a simulation is more to demonstrate how this model could be used given the information we have determined–carbon fluxes at fixed time intervals in the light and dark.  This was inspired by the previous studies of Belnap *et al* (2003) and Reed *et al* (2012), who showed that wetting frequency and season affect the carbon balance and production of pigments for radiation-protection. A follow-up study could explore this day/night ration and wetting frequency across seasons.

**Reviewer:** The authors should clarify whether or not upper and lower flux bounds are constrained by measurements when predicting flux through biopolymer reactions in the dark (Table 3). How well is CO2 production and biopolymer consumption predicted without constraining flux boundaries with measurements?

**Authors:** Upper and lower flux bounds were constrained by the average flux measurement in the dark. The description of Table 3 has been modified to include this clarification.
Using an unconstrained flux boundary for resources (represented as positive and negative infinity for the upper and lower bound) would result in an unconstrained/infinite $CO_2$ flux. The purpose of constraining the FBA problem is to reduce the degrees of freedom to solve for a desired unknown flux.
*Revision:*
For the final version of the description for Table 3, see Reviewer #2 manuscript revisions.

**Reviewer:** It is curious that M. vaginatus grows so slowly (weeks) and appears to miss vital reactions. Is it possible that it relies on near-obligate mutualistic partners in nature?

**Authors:** *M. vaginatus* certainly has mutualistic partners in nature (and in culture) that likely facilitate more rapid growth, but they are not essential for survival because M. vaginatus may be grown axenic. The naturally slow growth is also thought to be an evolved mechanism that allows survival in harsh arid environments. The "missing" of vital reactions is more a result of annotation, where key genes in M. vaginatus lack known homologs for functional assignment.

**Reviewer:** Please also share the metabolic model as an SBML file. Not everyone has access to Matlab.

**Authors:** An SBML file has been added to the supplementary information.

**Reviewer:** l. 67: a terrestrial cyanobacteria -> a terrestrial cyanobacterium l. 249: which a possible reason -> which is a possible reason l. 264: there is either a 12-hour wetting event -> or?

**Authors:** 67 and 249 have been corrected
264: where for each day there is either a light or dark 12-hour wetting event.
*Revisions (modified or added text is underlined):*
L67: "a terrestrial cyanobacterium"
L249: "which is a possible reason"
L264: "there is either a light or dark 12-hour wetting event"

22 Nov 2017

**Reviewer:** The SBML file generates a couple of warnings when loaded with R package sybilSBML. More specifically, two reactions appear to miss an identifier (no_id) and another reaction with identifier rxn00062 occurs twice. In addition, the number of genes and metabolites displayed in R and Matlab (with iNJ1153.mat) is not the same. I hope the authors can look into these issues before publication.

**Authors:** Thank you for your response and apologies for the issues regarding the SBML file. Please find attached a version that should run without errors in R. Modifications were minor. The number of unique genes does not reflect the number of total genes in the model due to redundancy in annotation, where more than one gene may be associated with the same function.

5 Dec 2017

**Reviewer:** The authors have addressed all my concerns.

**Authors:** Thank you for your detailed questions and suggestions. We have amended the manuscript with the changes mentioned in previous author comments.
*Revision: SBML file added to final manuscript supplementary information.*

**Reviewer #2**
30 Jan 2018

**Reviewer:** Flux balance modeling to predict bacterial survival during pulsed activity events by Jose et al. This is a very interesting study where genome information is combined with detailed measurements of storage products to reach the quantitative conclusion about the cost of activity in the dark and the light. Love the paper, well written, full of new information (for me). Not being familiar with the details of genome and flux balance approaches, my questions are mostly related to clarification.

**Authors:** Thank you for your detailed review, insightful comments and helpful suggestions. We have included below our responses and suggested clarifications to the manuscript.

**Reviewer:** In the metabolic model, I assume all growth functions, such as RNA, and protein production are included? Does the model assume that these other functions, such as protein production etc, happen at similar rates during the day as during the night?

**Authors:** Yes, production of RNA and protein are also included. The rate of production is related to the intensity of light, so that these rates are different during the day and night.

**Reviewer**:  The organisms were cultured at 4.5-10 umol m-2 s-1 (L83). How does this compare to the desert light levels, and how would an increase in light level affect the conclusion of this paper regarding wet/dry cycles during day/night? Is the light level PAR or total radiation?

**Authors**: While the radiation incident on the crust surface can be orders of magnitude higher that the one we used, it is subject to intense multiple scattering losses, so that only 1 percent of incident radiation remains a couple of millimeters down into the soil.  *M. vaginatus* makes a living within this steep light gradient, usually in the subsurface, coming up to the surface only when light intensity is very moderate, (morning, overcast conditions) and it has a "shade plant" phenotype (low

photosynthesis saturation intensity, heavy complement of light harvesting pigments and no sunscreen pigments). Isolate *M. vaginatus* does not grow in liquid media under desert light levels that we used previously to mimic more natural conditions in the lab with intact biocrusts (~600 umol m-2 s-1; DOI: 10.1038/ismej.2013.83). We conducted our experiments close to the growth optimum of *M. vaginatus* in liquid media. The light level given is PAR. This text has been added to the manuscript in the methods section (section 2.2)

*Revision*
L85: While the radiation incident on the crust surface can be orders of magnitude higher than the one we used, it is subject to intense multiple scattering losses, so that only 1 percent of incident radiation remains a couple of millimeters down into the soil. M. vaginatus makes a living within this steep light gradient, usually in the sub-surface, coming up to the surface only when light intensity is very moderate (in morning, overcast conditions). It has a "shade plant" phenotype possessing low photosynthesis saturation intensity, heavy complement of light harvesting pigments and no sunscreen pigments. Isolate M. vaginatus does not grow in liquid media under desert light levels that we used previously to mimic more natural conditions in the lab with intact biocrusts (~600 µmol m-2 s-1) (Rajeev et al., 2013). The growth conditions used are close to the growth optimum of M. vaginatus in liquid media.

**Reviewer:** Related topic, in the results and discussion (L 198, 199; Table 3) two light levels are defined. It was not clear when reading the paper what this meant, whether this was caused by a change in biomass production or changed light levels. Please clarify.

**Authors:** Thanks for pointing this out. The manuscript has been changed to clarify this.

*Revision*
L83: Because different culture containers were used between respiration experiments and biopolymer experiments, they possessed different photon fluxes (measured in µmol g$^{-1}$ h$^{-1}$), which is reflected in Table 3. This is accounted for in simulations.

**Reviewer:** L 116: please clarify why crotonic acid was determined. As far as I know, it is not mentioned in the results and discussion but seems to be the breakdown product of polyhydroxybutyrate. (?)

**Authors:** The manuscript has been changed to clarify this.

*Revisions*
Section 2.6 Title: "Quantification of PHB via LC/MS Measurement of Crotonic Acid"
L117: "Polyhydroxybutyrate breaks down to form crotonic acid in strongly acidic environments. The quantity of crotonic acid formed is used to calculate PHB quantity."

**Reviewer:** As a clarification, please explain why biopolymer reactions are included in the model, but not found in the genome. Similar for the other processes.

**Authors**: These clarifications have been added to the manuscript.

*Revisions*
L133: "Automatic annotations were further refined through manual annotation. This was necessary because automated modelling databases occasionally do not contain strong homologs for a certain function, and thus fail to assign it to the genome via simple homology search algorithms."

L144: "...reactions for the synthesis and metabolism of biopolymers polyphosphate, glycogen and PHB, which were not given within the automated annotation, were added after being found in the genome through extensive manual annotation."

L165: "Automated model generation typically does not assume that biopolymers act as resources that may accumulate or deplete."

**Reviewer:** L 150: how were LB and UB determined. On the one hand they seem to be the product of the model (Table 3) but at the same time constrain the model (L 150).

**Authors:** In Table 3 measured UB and LB refer to the experimental bounds, i.e. the standard deviation.
Modelled UB and LB refer to the upper and lower values obtained from sensitivity analysis. In modelled constraint reactions for light, UB and LB represent the assumed deviation that is input into sensitivity analysis.

The reaction flux UB and LB used as inputs in the model are fixed at the experimentally measured values (see L176, eqn 6). In sensitivity analysis, this is varied over the estimated deviation (See L182-186).

Due to this confusion, the LB and UB in Table 3 has been amended as seen below.

*Revision*
**"Table 3: Experimental and modelled flux values over light and dark conditions. Constraint fluxes are noted with a "*". Negative and positive $CO_2$ fluxes represent uptake and respiration respectively, while negative and positive biopolymer flux rates represent depletion and accumulation respectively. Measured "-" and "+" refer to the standard deviation. Modeled "-" and "+" refer to the upper and lower values obtained from sensitivity analysis. In modeled constraint reactions for light, "-" and "+" represent the assumed deviation that is input into sensitivity analysis."**

| | Measured | | | Modeled | | |
|---|---|---|---|---|---|---|
| | Flux ($\mu$mol g$^{-1}$ h$^{-1}$) | - | + | Flux ($\mu$mol g$^{-1}$ h$^{-1}$) | - | + |
| **Light (1)** | | | | | | |
| Light* | 141 | | | 141 | 93 | 211 |
| $CO_2$ | -19.6 | -23.3 | -15.9 | -33.4 | -50.3 | -21.8 |
| **Light (2)** | | | | | | |
| Light* | 4670 | | | 4670 | 3551 | 6290 |
| Glycogen | 9.99 | 9.47 | 10.51 | 3.07 | 0.00 | 9.30 |
| PHB | 0.0366 | 0.0050 | 0.0683 | 0.121 | 0.044 | 0.250 |
| Polyphosphate | 2.28 | 1.57 | 2.99 | 2.82 | 1.00 | 5.70 |
| **Dark** | | | | | | |
| Light* | 0.00 | | | 0.00 | | |
| Glycogen* | -2.49 | -6.08 | 1.09 | -2.49 | -9.66 | 4.67 |
| PHB* | -0.14 | -0.17 | -0.11 | -0.136 | -0.20 | -0.07 |
| Polyphosphate* | -0.02 | -1.27 | 1.22 | -0.02 | -2.51 | 2.46 |
| $CO_2$ | 17.3 | 14.1 | 20.6 | 11.9 | 0 | 48.5 |

**Reviewer:** Table 1: what is the relevance of the reactions mentioned such as hydrogen production

in the table, but neither curated not found in the genome.

**Authors:** Fermentation and nitrogen fixation are known to be important pathways in biocrust metabolism. We annotated these to investigate if *M. vaginatus* could play a key role. Hydrogen production was annotated because hydrogen evolution was measured from crusts in previous experiments, though not reported.

**Reviewer:** L 170-173 and Table 2: I am not familiar with flux balance calculations, so the phrase PHB -> nothing is confusing? Please add explanation in one additional sentence.

**Authors:** This has been changed.

*Revision:*
L167: "…where $A$ is a side reactant, $B$ is a side product, $X$ is a polymer subunit, and $X_n$ is a polymer with length $n$. 'Nothing" is not a physical term, but a mathematical way to describe resource accumulation in a steady state simulation."

**Reviewer:** Table 3 and Fig. 2c seem to have some overlap.

**Authors:** This is true, they are just different representations of the same data, we've decided it best to keep them because they may be useful to different types of readers. While the graph provides a visual comparison of experiment and model predictions, the tabulation of values may be helpful for those performing future simulations and studies.

**Reviewer:** L 214: please elaborate how polyP can be used in other ways than an energy source

**Authors:** This has been corrected as described below:
*Revision*
L213-214: "Polyphosphate is likely an important biopolymer for *M. vaginatus* across many different stressed conditions as a reservoir of phosphate for later growth, though so-called "luxury uptake" and storage when growth is halted by some other factor, and as a reservoir of energy in the form of phosphate-phosphate bonds under conditions of abundant energy generation, phosphate and a lack of conditions to use it for growth or homeostasis. This importance of polyphosphate has been identified in gene expression studies (Rajeev et al., 2013)."

**Reviewer:** L 218: change consistent to constant or words similar to that.

**Authors:** Done

*Revision*
L218: "…storage polymer or for other metabolic activities that do not require a constant energy source, such as replication."

**Reviewer:** L 52: diel does not need to be capitalized

**Authors:** Done

*Revision*
L52: "During the diel cycle…"

**Reviewer:** L 63: the use of the phrases dark and light reactions are confusing: they have very

specific meaning in the study of photosynthesis, but I don't think that is what is meant here. Please Ăn replace with something like metabolism in the dark versus in the light.

**Author**: Good point. We avoid those terms now in the manuscript to avoid confusion.

*Revision*
L63: "Therefore, we expect wet-up and dry-down metabolism in the dark likely have fixed biopolymer costs whereas metabolism in the light enables replenishment of biopolymer reserves."

**Reviewer:** L71: complex sentence that can be simplified.

**Author:** Done

*Revision:*
L70-71: "We interpret these results using a simple cost/benefits framework. The "cost" is biopolymer depletion in the dark, and the "benefit" is biopolymer accumulation in the light.

**Reviewer:** L 94: add rcf to the list of abbreviations, and add units

**Author:** Done. Changed to "x g" (g-force). All units will be added to the abbreviations.

*Revisions:*
Abbreviations (L284):
| GPR | Gene-Protein-Reaction |
| PAR | Photosynthetically Active Radiation |

Units
| Å | Angstrom |
| g | gram |
| h | hour |
| L | liter |
| m | meter |
| M | moles per liter |
| mg | milligram |
| min | minute |
| mL | milliliter |
| mm | millimetre |
| mmol | millimole |
| mol | mole |
| rpm | rotations per minute |
| s | second |
| x g | g-force (x $9.81 \ m \ s^{-2}$) |
| µg | microgram |
| µL | microliter |
| µm | micrometer |
| µmol | micromole |

**Reviewer:** L 134: what are GPR relations

**Author:** The Gene-Protein-Reaction relation. It is a standardized description of the link between a gene, its associated protein, and the associated reaction in genome-scale metabolic models. We include this in abbreviations.

*Revision*
Abbreviations (L284):
GPR            Gene-Protein-Reaction
PAR            Photosynthetically Active Radiation

**Reviewer:** L 135: comma after databases can be removed

**Author:** Done
*Revision*
L135: "…analysis, and the databases KEGG and MetaCyc…"

**Reviewer:** L 168: why the word "side" with reactant and product?

**Author:** The reactants and products of interest are the biopolymer and its subunit. A and B are used to generalize other chemicals involved in the process.

**Reviewer:** L 194: change profiles to concentrations

**Author:** Done
*Revision*
L194: "…carbon dioxide profiles concentrations varied linearly with time…"

**Reviewer:** L 246: Add year after reference (Knoop)

**Author:** Done
*Revision*
L246: "(Knoop *et al.*, 2013)"

**Reviewer:** L 247: is the efficiency measured at the same light level?

**Author:** No, the rates are normalized to the photon intensity in this comparison.

[revised manuscript text omitted]

713-724, 2003.
445

[Figure]

**Figure 1: Metabolic reconstruction process diagram.**

450

[Figure]

<table>
<tr><td>(a)</td><td>(b)</td><td>(c)</td></tr>
</table>

**Figure 2: Comparisons of experimental and modelled CO₂ accumulation (a) in the light, where a negative value indicates uptake, and (b) in the dark, where a positive value indicates respiration. Biopolymer and CO₂ flux rates are compared in (c), where error bars on modelled flux rates are the upper and lower bounds determined through sensitivity analysis; error bars on measured flux rates are standard deviations.**

[Figure]

**Figure 3: Predicted glycogen depletion over 1 month under three climate scenarios, where the ratio of light:dark wetting events are 1:1, 1:3, and 1:5.**

**Table 1: Comparison of curation and automated annotation.**

| Major manually curated pathways | Predicted in curation | Predicted in RAST annotation |
|---|:---:|:---:|
| All amino acid biosynthetic pathways | x | x |
| Ammonium Assimilation | x | x |
| Bifidobacterium Shunt | x | x |
| Heterolactic Acid Fermentation | | |
| Homolactic Acid Fermentation | x | x |
| Mixed Acid Fermentation | | |
| Nucleoside triphosphate biosynthetic pathways | x | x |
| Photosynthetic light reactions | x | |
| Calvin Cycle | x | x |
| Nitrate Assimilation | x | |
| Nitrogen Fixation | | |
| Glycogen Biosynthesis | x | |
| Glycolysis | x | x |
| Hydrogen Production | | |
| Pentose Phosphate Cycle | x | x |
| Sulfur and Sulfate Reduction | x | x |
| TCA Cycle | x | x |
| Peptidoglycan Biosynthesis | x | x |
| β-polyhydroxybutyrate synthesis | x | |
| Cyanophycin synthesis | x | |
| Polyphosphate synthesis | x | |

465 **Table 2: Modelled Biopolymer Reactions.**

| Reaction | Description |
|---|---|
| **ATP ⬌ADP + Polyphosphate** | Polyphosphate synthesis/degradation |
| **Polyphosphate ⬌ Nothing** | Polyphosphate sink |
| **Glucose-1-phosphate ← Phosphate + H(+) + Glycogen** | Glycogen degradation |
| **ADP-Glucose → Glycogen + ADP** | Glycogen synthesis |
| **Glycogen ⬌Nothing** | Glycogen Sink |
| **(R)-3-Hydroxybutanoyl-CoA ⬌ CoA+PHB** | PHB synthesis/degradation |
| **PHB⬌Nothing** | PHB sink |

**Table 3: Experimental and modelled flux values over light and dark conditions. Constraint fluxes are noted with a "*". Negative and positive CO$_2$ fluxes represent uptake and respiration respectively, while negative and positive biopolymer flux rates represent depletion and accumulation respectively. Measured "-" and "+" refer to the standard deviation. Modelled "-" and "+" refer to the upper and lower values obtained from sensitivity analysis. In modelled constraint reactions for light, "-" and "+" represent the assumed deviation that is input into sensitivity analysis.**

| | Measured | | | Modeled | | |
|---|---|---|---|---|---|---|
| | Flux (μmol g$^{-1}$ h$^{-1}$) | LB- | UB+ | Flux (μmol g$^{-1}$ h$^{-1}$) | LB- | UB+ |
| **Light (1)** | | | | | | |
| Light* | 141 | | | 141 | 93 | 211 |
| CO$_2$ | -19.6 | -23.3 | -15.9 | -33.4 | -50.3 | -21.8 |
| **Light (2)** | | | | | | |
| Light* | 4670 | | | 4670 | 3551 | 6290 |
| Glycogen | 9.99 | 9.47 | 10.51 | 3.07 | 0.00 | 9.30 |
| PHB | 0.0366 | 0.0050 | 0.0683 | 0.121 | 0.044 | 0.250 |
| Polyphosphate | 2.28 | 1.57 | 2.99 | 2.82 | 1.00 | 5.70 |
| **Dark** | | | | | | |
| Light* | 0.00 | | | 0.00 | | |
| Glycogen* | -2.49 | -6.08 | 1.09 | -2.49 | -9.66 | 4.67 |
| PHB* | -0.14 | -0.17 | -0.11 | -0.136 | -0.20 | -0.07 |
| Polyphosphate* | -0.02 | -1.27 | 1.22 | -0.02 | -2.51 | 2.46 |
| CO$_2$ | 17.3 | 14.1 | 20.6 | 11.9 | 0 | 48.5 |

470

475